# Hurdles of Sperm Success: Exploring the Role of DNases

**DOI:** 10.3390/ijms26146789

**Published:** 2025-07-15

**Authors:** Jaime Gosálvez, Carmen López-Fernández, Javier Bartolomé-Nebreda, Carlos García de la Vega

**Affiliations:** 1Unidad de Genética, Departamento de Biología, Universidad Autónoma de Madrid, 28049 Madrid, Spain; carlos.delavega@uam.es; 2Departamento de Investigación y Desarrollo, Halotech DNA S.L., Parque Científico de Madrid, 28049 Madrid, Spain; carmen.lopez@halotechdna.com (C.L.-F.); javier.bartolome@halotechdna.com (J.B.-N.)

**Keywords:** sperm DNA fragmentation, male factor, infertility, DNase activity

## Abstract

The incidence of sperm DNA fragmentation (SDF) in the ejaculate has garnered increasing attention in recent years due to its negative impact on reproductive outcomes. SDF involves two primary types of damage to the canonical double helix of DNA: single-strand breaks and double-strand breaks. Both of these can occur throughout the entire process of gametogenesis. Determining the precise causes of elevated SDF remains challenging, as it is influenced by a wide range of physiological processes and environmental factors. This review comprehensively explores the mechanisms underlying SDF, with a particular emphasis on the critical role of deoxyribonucleases (DNases) across different stages of male gamete development, as well as their relevance in assisted reproductive technologies (ART).

## 1. Introduction

The presence of spermatozoa harboring fragmented DNA within the ejaculate is a well-documented phenomenon, and when elevated, constitutes a significant factor contributing to male infertility [1]. Sperm DNA fragmentation (SDF) encompasses two principal types of structural disruption in the canonical DNA double helix: (i) single-strand breaks (SSBs) and (ii) double-strand breaks (DSBs) [2]. SDF is frequently observed in various clinical conditions associated with infertility, suggesting a multifactorial etiology. Thus, accurately identifying the specific causes of elevated SDF remains challenging [3,4].

A critical question thus emerges: At what stage(s) of gametogenesis do these molecular alterations arise, and what are the underlying mechanisms responsible for these modifications affecting the DNA molecule? SDF can originate at four key stages during the development and processing of male gametes: (i) the mitotic expansion of spermatogonial stem cells, (ii) meiotic division, (iii) histone-to-protamine replacement, and (iv) post-ejaculatory sperm handling.

Spermatogenesis comprises sperm development within the seminiferous tubules of the testes, encompassing chromatin remodeling and capacitation. This process begins during puberty and continues throughout the reproductive lifespan, converting spermatogonial stem cells into immature spermatozoa. This culminates in the formation of round spermatids following meiosis. In humans, the estimated duration of spermatogenesis ranges from 64 to 74 days, whereas in rodent models it is significantly shorter [5].

Spermatogenesis can be divided into three phases: (i) The proliferation of spermatogonia through mitosis, yielding primary diploid spermatocytes. (ii) The meiotic phase in which the primary spermatocytes undergo meiosis I to form haploid secondary spermatocytes, which subsequently divide single chromosomes (meiosis II) to generate round haploid spermatids. These spermatids maintain a single chromatid per chromosome and do not replicate DNA. (iii) The spermiogenesis that involves a morphogenetic transformation in which round spermatids differentiate into structurally mature but functionally immature spermatozoa. A series of intricate cytological changes are produced, including condensation of the haploid genome, acrosome biogenesis, flagellum formation, cytoplasmic reduction, and mitochondrial reorganization. These processes are essential for endowing spermatozoa with the structural features necessary for fertilization. Spermatozoa generated at this stage are non-motile and require further functional maturation during the epididymal transit. The duration of spermiogenesis in humans is approximately 23–24 days [6], whereas in murine models, it is completed within seven days [7].

Finally, during natural fertilization, sperms undergo final maturation as they traverse the female reproductive tract. However, in ART, including intrauterine insemination (IUI), in vitro fertilization (IVF), and intracytoplasmic sperm injection (ICSI), sperm are subjected to artificial environments and manipulative stressors. These interventions may exacerbate DNA damage, a phenomenon known as iatrogenic damage [8,9].

Within this framework, four critical junctures where sperm DNA integrity may be compromised via strand breakage and subsequent repair or degradation are recognized. These include (i) mitotic errors and repair deficiencies during spermatogonia proliferation, (ii) programmed DSBs and recombination during meiosis, (iii) chromatin restructuring during histone-to-protamine exchange during spermiogenesis, and (iv) exogenous insults during the in vitro manipulation of sperm in ART protocols.

While it is well established that various exogenous and endogenous stressors, including oxidative stress, thermal exposure, and genomic instability, can induce SDF, this review focuses specifically on the role of deoxyribonucleases (DNases) in mediating DNA fragmentation across these four developmental stages. Because of its importance in ART, emphasis is placed on elucidating the mechanisms of DNase-mediated DNA cleavage, particularly in relation to iatrogenic DNA damage and the presence of DNases in the seminal plasma.

## 2. Sperm Degradation Along Gametogenesis

Two distinct mechanisms of DNA breakage have been identified during gametogenesis. The first involves the essential modifications required for functional processes such as DNA repair, meiotic recombination, and histone-to-protamine transition. The second set of mechanisms facilitates the removal of circulating free DNA or DNA residing in compromised cells, which may arise from errors in the preceding processes or damage induced by external factors. Apoptosis and necrosis are the two primary mechanisms responsible for the elimination of abnormal cells. The various processes of DNA cleavage that occur throughout gametogenesis involve distinct enzymatic activities. Figure 1 highlights the primary enzymatic activity associated with each specific stage, while Table 1 provides a summary of the main enzymes and enzymatic pathways participating in DNA cleavage at different stages of gametogenesis.

## 3. Apoptotic Mechanisms in Sperm Death

Apoptosis plays a pivotal role in the selective elimination of defective germ cells during spermatogenesis and, possibly, after ejaculation in the female reproductive tract. This tightly regulated mechanism serves as a safeguard against the persistence of spermatozoa with DNA damage, chromosomal anomalies, or structural abnormalities. Apoptosis maintains testicular homeostasis and minimizes cellular competition for limited nutritional and hormonal resources by regulating the removal of damaged or superfluous germ cells [23].

In the male germline, apoptotic activity has been observed as early as the fetal developmental stage [24]. During spermatogenesis, apoptosis affects spermatogonia [25,26], meiotic spermatocytes [27], and occasionally post-meiotic cells. This process is initiated in response to various stressors, including hormonal imbalances, infections, exposure to environmental toxins, and oxidative stress [28,29], which influence all stages of germ cell differentiation.

One of the earliest molecular events in apoptosis is mitochondrial outer membrane permeabilization, which facilitates the release of cytochrome c into cytosol. Under typical apoptotic conditions, this release leads to the assembly of the apoptosome (complex formed by cytochrome c and apoptosis-activating factor) and the subsequent activation of the caspase cascade, primarily involving initiator caspases, such as caspase-8 and caspase-9. A key hallmark of early-stage apoptosis is the externalization of phosphatidylserine (PS). PS is a phospholipid normally confined to the inner leaflet of the plasma membrane. PS also binds to proteins such as annexin V, which is a calcium-dependent protein with a high affinity for PS [30]. Although PS externalization is not directly involved in DNA fragmentation, it typically occurs concomitantly during the early apoptosis execution phase. In the testicular environment, externalized PS serves as a signal for Sertoli cells, facilitating the phagocytic removal of defective germ cells [31].

Two primary forms of apoptosis have been characterized in the testicular environment: complete and abortive. Complete apoptosis entails full execution of the cell death program and is commonly observed in mitotically active spermatogonia and primary spermatocytes. During meiotic prophase I, it results from the activation of the corresponding quality checkpoint. In contrast, abortive apoptosis refers to the initiation of the apoptotic pathway without completion of the cascade, leading to cells with compromised functionality. This incomplete process, which has been extensively studied in oncology, neurodegeneration, and immunology, has also been observed in spermiogenesis [32]. Abortive apoptosis becomes particularly prominent following meiosis during the transformation of round spermatids into spermatozoa. Cells undergoing this process may evade immediate phagocytosis and can persist as residual bodies or malformed spermatids, which can be cleared by Sertoli cells [33] or ejaculated. The role of apoptosis in mature ejaculated spermatozoa remains debatable. This topic warrants further discussion, particularly owing to its potential implications in the context of ART, as explored in subsequent sections.

Caspases are the most prominent enzymes involved in DNA degradation during apoptosis. However, caspases do not directly degrade DNA; instead, they activate the nucleases responsible for DNA fragmentation. One of the key nucleases in this process is Caspase-Activated DNase (CAD), an endonuclease that cleaves chromosomal DNA into nucleosomal units (~180–200 bp fragments), a hallmark of apoptosis [18,19]. Under normal conditions, CAD remains inactive due to its association with ICAD (an Inhibitor of CAD). However, Caspase-3 and Caspase-7 can cleave ICAD, releasing active CAD, which subsequently translocates to the nucleus to degrade chromosomal DNA.

In abortive apoptosis, the apoptotic cascade is initiated but fails to proceed through full execution and the downstream activation of effector caspases, particularly caspase-3, is frequently attenuated or fails to occur entirely. This incomplete activation has been linked to a reduction in the intratesticular testosterone concentration, which is known to influence caspase-3 activity [34]. Moreover, the presence of singlet oxygen, in the context of oxidative stress, inhibits the enzymatic activity of caspase-9 and caspase-3, thereby contributing to the suppression of apoptosis [35]. Additional anti-apoptotic mechanisms may also intervene, including the overexpression of Bcl-2 (B-cell lymphoma 2) family proteins, the activation of pro-survival pathways, such as PI3K (Phosphoinositide 3-kinase)/AKT, and the expression of c-FLIP (cellular FLICE-inhibitory protein), which inhibits caspase-8 activation [36,37]. These factors collectively arrest the apoptotic cascade, allowing cells to survive despite the initiation of pro-apoptotic signals. In summary, this truncated apoptosis pathway after caspase activity may contribute to the persistence of SDF in the ejaculate.

### 3.1. DNA Cleavage in Spermatogonia

During spermatogonial proliferation, DNA repair mechanisms are essential for preserving genome integrity, which is critical for male fertility and the faithful transmission of genetic information to the next generation. As diploid somatic cells, spermatogonia utilize DNA repair pathways that are largely conserved among other somatic cell types. The initial line of defense involves several fundamental DNA repair mechanisms that counteract endogenous and exogenous sources of DNA damage.

Some of them permit the healing of distinct failures in DNA single strands, namely small base lesions and mismatches. This is the case for BER (Base Excision Repair), MMR (Mismatch Repair) and NER (Nucleotide Excision Repair). Importantly, the pathways for resolving these errors include the participation of some DNases that transiently promote DNA breakages. Consequently, whenever these breakages are not properly resolved, sperm may show variable rates of DNA damage.

With regard to DSBs produced by distinct DNases, two major repair pathways must be considered: (i) Homologous Recombination (HR) is a high-fidelity mechanism for the repair of DSBs and is primarily active during the S and G2 phases of the cell cycle. In proliferating spermatogonia, HR plays a key role in maintaining genomic stability prior to meiosis onset. (ii) Non-Homologous End Joining (NHEJ) provides a rapid, although error-prone, means of repairing DSBs and is active throughout the cell cycle. While this pathway is necessary for genome maintenance in the early stages of spermatogonia, excessive reliance on NHEJ can lead to mutagenesis and is therefore tightly regulated.

The utilization of various DNA repair mechanisms is highly dependent on the type of lesion incurred and their effectiveness is influenced by the age of the individual [38,39,40]. It is estimated that up to 75% of germ cells are lost during the spermatogonial stage. This pool of cells includes those bearing distinct DNA errors, including the accidental occurrence of DSBs [41]. In *Drosophila*, Lu and Yamashita [42] showed that every spermatogonia within a cyst, where only some of them exhibit DNA damage, enters synchronously in an elimination pathway. Moreover, the surveillance of DNA damage was also tested in the spermatogonial stem cells of mice. In this case, the induced DSBs were repaired via the NHEJ pathway. Failures in the repair process lead to the elimination of cells through apoptosis, using Caspase-3 and PARP-1 (Poly (ADP-ribose) polymerase-1) as markers [43].

When these repair pathways fail to resolve DNA damage, the cells may undergo programmed cell death via apoptosis. Under severe or unregulated damage conditions, necrosis may occur to eliminate compromised cells and preserve tissue homeostasis (see below).

### 3.2. DNA Cleavage During Spermatogenesis

Meiosis is the specialized form of cell division that underpins sexual reproduction and leads to gamete formation. This reductional division of chromosomes involves prolonged prophase I, during which the homologous chromosomes undergo synapsis and recombination. A key feature of this process is the early and programmed induction of DSBs along chromosomes, primarily mediated by protein SPO11 (Sporulation protein 11) in humans (Spo11 in mice) [13]. SPO11 is an evolutionarily conserved protein closely related to Topoisomerase VI, a protein described in Archaea [44]. Its primary function is to induce DNA cleavage in one chromatid of a homologous chromosome. SPO11 remains covalently bound to the broken DNA ends. Other proteins are needed for the removal of SPO11 and DNA repair. This process culminates in the synapsis of homologues and, at specific loci, recombination events [45].

Defects in SPO11 function are consistently associated with meiotic arrest during early prophase I, which leads to disrupted spermatogenesis and impaired testicular development. Consequently, sterility is observed in both mice and humans [46,47,48,49,50]. The absence of programmed DSBs disrupts the recombination process, thereby preventing the formation of the “obligatory chiasma” necessary for the proper segregation of homologs during the first meiotic anaphase [51]. Notably, there are chromosomal regions in which SPO11 does not induce cleavage, including telomeres, centromeres, and nucleolar organizers [45]. In males, SPO11 activity is suppressed in the non-pseudoautosomal region (non-PAR) of the sex chromosomes. DSBs are almost exclusively generated in the homologous regions of the X and Y chromosomes (i.e., in the PAR). In mice, Spo11 activity is regulated by ATM (Ataxia-telangiectasia mutated) kinase, and the suppression of ATM activity leads to a significant increase in DSBs in non-PAR regions [52]. Furthermore, ANKRD3 (Protein-Serine-Threonine Kinases), a protein involved in the initiation of meiotic synapsis, is essential for recombination within PAR of the X and Y chromosomes. Its absence has been associated with male sterility in mice [53]. Mice deficient in Spo11 exhibit diminished seminiferous tubules and a complete absence of spermatids. The remaining cells within the tubules displayed fragmented, highly condensed chromatin, indicative of an apoptotic process, as confirmed by TUNEL (Terminal deoxinucleotidil transferasa dUTP nick-end labeling) assays [46,54].

### 3.3. DNA Cleavage During Spermiogenesis

Upon completion of male meiosis, spermatids undergo a dramatic reorganization of their haploid genomes. This transformation is driven by the incorporation of protamine-specialized basic proteins that replace histones in most of the chromatin. For instance, in human sperm, only 15% of the chromatin retains its nucleosomal organization [55]. The sequence of protein replacement has been extensively studied, and the process, at present, is almost fully understood [56]. Initially, various histone variants and modifications contribute to altering the nucleosomal structure. Subsequently, transition proteins (TNPs) facilitate the replacement of histones with protamines (PRM1 and PRM2 in humans) [57,58]. This results in a chromatin structure adopting a more compact toroidal conformation. In fact, the sperm nucleus is approximately 40 times smaller than any of the somatic line cells [55]. While histone octamers regularly wrap 147 base pairs of DNA in every nucleosome, protamines are positioned every 10–15 base pairs in the toroids of the sperm cells.

The exchange of DNA-interacting basic proteins necessitates the generation of DSBs despite the absence of repair templates in spermatid chromosomes [59,60]. Two mechanisms have been proposed for the induction of DSBs in sperm nuclei: (i) torsional stress induced by the replacement of the nucleosomal structure, and (ii) the action of topoisomerases [61]. While torsional stress could result from the accumulation of DNA supercoils as histones are replaced, this alone may not fully explain the presence of 3′OH DNA ends observed in sperm nuclei by the TUNEL assay [61]. An alternative explanation involves the activity of topoisomerases, particularly TOPO2B (Topoisomerase 2β), which has been specifically detected during sperm chromatin remodeling in mice [17]. Similarly, in the ciliate *Tetrahymena thermophila*, Top2G (evolutionarily related to TOPO2B) is active in gametic nuclei. Subsequently, γH2AX (γHistone 2AX) foci are detected, thus marking the locations of DSBs. Interestingly, when Top2G is depleted, nuclei do not exhibit γH2AX labelling [62]. These authors also presented relevant results indicating that SPO11 also contributes to the appearance of DSBs in post-meiotic stages. Gouraud et al. [61] had already proposed this role for SPO11 in spermatids of mice since this protein is also expressed in that stage. In summary, both TOPO2B and SPO11 may contribute to the DSBs that are inherent to the remodeling of sperm chromatin. Since sperm chromosomes lack a repair template, DSB repair likely follows an alternative NHEJ pathway as, at least, two key proteins [DNA-PKcs (DNA-dependent Protein Kinase catalytic subunit) and KU] involved in the classical NHEJ are not expressed in spermatids [63].

During this critical phase of chromatin remodeling, any failure in the activity of either TOPO2B or SPO11, or any defect in subsequent repair pathways, would lead to improperly condensed sperm chromatin. In such cases, apoptotic mechanisms are thought to eliminate the affected spermatids [61].

### 3.4. DNA Cleavage in Ejaculated Sperm

The presence of apoptotic markers in sperm and their role in sperm DNA degradation is open to debate. Apoptotic signals generated during chromatin remodeling remain and can be visualized after ejaculation in sperm [64,65]. However, evidence suggests that their inability to undergo apoptosis is linked to the absence of cytoplasm. Supporting this notion, sperm selection via swim-up techniques (used to isolate mature sperm), followed by the incubation of the recovered fraction at 37 °C for up to 24 h, did not show an increase in phosphatidylserine translocation or DNA fragmentation, indicating that spermatozoa are incapable of initiating apoptosis, at least under those in vitro conditions [66]. Moreover, fibroblast-associated death receptor activation had no functional relevance in mediating caspase activation in human ejaculated spermatozoa [67].

In contrast, the in vitro exposure of ejaculated spermatozoa to bacteria correlates with a decline in the proportion of sperm exhibiting normal mitochondrial transmembrane potential and an increase in phosphatidylserine externalization, both hallmarks of apoptosis. Additionally, these spermatozoa show TUNEL positivity, which is a marker of DNA damage [68,69]. Further evidence suggests that apoptosis is an active pathway that can be induced in ejaculated sperms. For instance, semen samples treated with apoptosis inducers such as betulinic acid or thapsigargin showed the significant activation of caspase-9 and caspase-3, along with a marked reduction in sperm motility [70].

From a biological standpoint, the presence of apoptotic mechanisms in ejaculated sperm appears to be plausible. However, a key question remains: is this process restricted to sperm that are not fully mature, as they may still retain residual cytoplasmic components necessary for apoptosis activation? Additional experimental evidence is required to resolve this issue.

If fully mature sperm lacks the capacity to undergo apoptosis, alternative mechanisms must ensure the removal of spermatozoa that fail to traverse the cervix in the female reproductive tract. Thus, post-ejaculatory sperm elimination serves critical functions, including the removal of low-quality sperm (i.e., sperm with DNA damage, motility defects, or structural abnormalities), and occurs through phagocytosis by the female immune system, particularly by macrophages.

Since the dysregulation of apoptosis, whether excessive or insufficient, can lead to male infertility, impaired sperm quality, and an increased risk of genetic abnormalities in offspring [71], alternative pathways, such as necrosis, which may be associated with pro-inflammatory responses, may play a predominant role in sperm clearance (see below).

### 3.5. Additional Enzymes Involved in DNA Cleavage During Gametogenesis

In addition to the physiological enzymatic processes associated with the pre-meiotic proliferation, meiotic synapsis and recombination, and chromatin remodeling during protamination, other nucleolytic pathways are engaged to eliminate defective sperm resulting from errors in the repairing pathways of the above processes, as well as from cumulative cellular insults.

Several nucleases implicated in apoptotic DNA fragmentation include Endonuclease G (EndoG) and Apoptosis-Inducing Factor (AIF). EndoG is a mitochondrial nuclease released into the cytosol following the permeabilization of the mitochondrial outer membrane. It contributes to the degradation of both nuclear and mitochondrial DNA (mtDNA) during apoptosis, particularly during the terminal phases of sperm maturation, selectively targeting defective spermatozoa that fail to migrate through the female reproductive tract [14,72]. However, its specific contribution to apoptotic DNA fragmentation remains a subject of debate, as some studies have reported limited or non-essential roles for EndoG in cell death and direct evidence of its involvement in ejaculated sperm necrosis remains scarce [15].

AIF is a mitochondrial flavoprotein that plays a central role in caspase-independent apoptosis [73]. During spermatogenesis, AIF is highly expressed in spermatogonia and primary spermatocytes, where it contributes to germ cell quality control by eliminating cells with DNA lesions or developmental defects [10]. In contrast, its expression declines markedly during the later stages of sperm maturation because of the lack of complete apoptotic machinery in mature spermatozoa. AIF mediates large-scale DNA breakage, producing fragments ranging from 50 kb to 1 Mb that are distinct from the oligonucleosomal pattern that is typical of caspase-dependent pathways [11,74].

Exonucleases of the TREX family, particularly Three Prime Repair Exonuclease 1 (TREX1), are involved in the clearance of apoptotic DNA from defective germ cells. These 3′–5′ exonucleases degrade both single- and double-stranded DNA fragments, including those generated by caspase-activated DNase (CAD), and act in concert with DNase II to complete the DNA digestion following apoptosis. Their activity is essential to prevent chronic inflammation, autoimmune responses, and testicular tissue damage [75]. The functional loss or genetic disruption of TREX enzymes is associated with infertility in mice (both males and females) [76]. TREX1 is particularly active during the early stages of spermatogenesis and has been detected in Sertoli cells, supporting its role in germ cell surveillance and debris clearance [12].

Recently, the role of Poldip2 (Polymerase delta-interacting protein 2), a mitochondrial exonuclease specifically expressed during the late stages of spermatogenesis, was characterized in the elimination of sperm mtDNA in *Drosophila* [16]. Moreover, the loss of Poldip2 disrupts mtDNA clearance in elongated spermatids and hinders the progression of the individualization complexes responsible for removing cytoplasmic components and organelles. Over time, sperms from Poldip2-deficient mutants exhibit pronounced nuclear genome fragmentation, resulting in male sterility [77].

It is increasingly evident that sperm cell death involves a more complex network of pathways than originally appreciated. In addition to classical apoptosis, alternative regulated cell death mechanisms, such as necroptosis, a programmed form of necrosis linked to pro-inflammatory signaling, and pyroptosis, governed by inflammasome activation, may play significant roles in various stages of germ cell development and elimination [78]. Ferroptosis, a recently described regulated cell death modality characterized by iron-dependent lipid peroxidation, has been implicated in germ cell death associated with oxidative stress and impaired redox homeostasis [79,80,81]. A more comprehensive understanding of these alternative or complementary cell death pathways is critical for elucidating their roles in sperm elimination and reproductive homeostasis.

## 4. Necrosis in Sperm Death

Unlike apoptosis, a tightly regulated and programmed mechanism of cell death, necrosis has traditionally been regarded as a chaotic and uncontrolled process of cellular destruction that affects all cellular components and is typically associated with an inflammatory response. However, recent evidence suggests that necrosis may be considered an alternative form of programmed cell death that operates through a pathway distinct from apoptosis [78,82,83].

While apoptosis serves as the primary quality control mechanism for the elimination of defective sperm, necrosis may also contribute to sperm elimination under certain extreme conditions such as exposure to elevated oxidative stress, inflammation, infections, or mechanical trauma during intercourse [66]. In mammals, the vaginal and cervical mucus traps sperm, preventing their progression and leading to necrotic demise. Most necrotic sperms are subsequently phagocytosed by macrophages and neutrophils, which are integral components of the innate immune response in the female reproductive tract. Certain immune cells, such as dendritic cells, may recognize sperms as foreign entities, triggering an additional inflammatory response and recruiting immune cells to the site [84]. This immune response is particularly relevant when sperm enters the cervical mucus or when minor tissue damage occurs during intercourse. Such immunological reactions may have adverse consequences on fertility, as they can lead to the development of anti-sperm antibodies, particularly in cases of repeated exposure or underlying immune dysregulation. These antibodies bind to sperm, marking them for destruction by immune cells, and in some cases may contribute to infertility in both men (via anti-sperm antibodies in semen) and women (through immune responses in the reproductive tract) [85,86].

Necrosis can occur at various stages of spermatogenesis, as well as while sperm transit the female reproductive tract. Although apoptosis is the predominant mechanism for the elimination of defective or excess germ cells, necrosis can also occur, particularly under pathological conditions or following exposure to toxic agents. For instance, studies have demonstrated that in certain animal models, immature germ cells, including spermatogonia, may undergo necrosis as a means of maintaining testicular homeostasis [87]. Necrosis was also observed during spermatogonial proliferation. In murine spermatogenesis, the initial wave of spermatogonial expansion via mitosis is accompanied by a pronounced apoptotic event that requires the activation of the TP53 gene, which encodes the tumor suppressor protein p53, a key regulator of cell cycle arrest [88]. Alternatively, the affected cells may be eliminated through a necrotic pathway [89]. Necrosis during meiosis or spermiogenesis has not been reported. However, once sperm enters the female reproductive tract, they encounter various environmental challenges, such as the acidic pH of the vagina, which is hostile to sperm and results in the rapid demise of many cells post-ejaculation.

Cell death is accompanied by membrane degradation and DNA release, which may act as reactive molecules. Free circulating DNA molecules can function as DAMPs (Damage-Associated Molecular Patterns), triggering an immune response [90]. DAMPs, including DNA, RNA, and proteins, are released from damaged or dying cells and serve as immunological danger signals by activating Pattern Recognition Receptors (PRRs) on immune cells. This interaction activates signaling pathways that promote inflammation and immune responses, potentially affecting various organ systems [91,92]. Additionally, mtDNA, which becomes highly accessible following sperm membrane degradation, may function as a DAMP [93].

Both apoptosis and necrosis were observed in ejaculated sperm. A study examining human ejaculated spermatozoa identified morphological and biochemical markers indicative of both types of cell death, suggesting that necrosis plays a role in the post-ejaculatory decline in sperm quality [66]. During necrosis, DNA degradation is facilitated by DNase I, an endonuclease active in both extracellular and intracellular environments that requires divalent cations (Ca^2+^ and Mg^2+^) for activity and is released from lysosomes upon cell lysis [20]. Additionally, DNase II, a lysosomal endonuclease that operates under acidic conditions such as those present in the vagina, plays a crucial role in degrading DNA from apoptotic and necrotic cells within macrophages. Unlike DNase I, DNase II does not require divalent cations for its enzymatic activity [21]. Other studies have highlighted the impact of inflammatory factors within the ejaculate, particularly the increased presence of proinflammatory cytokines such as tumor necrosis factor-alpha (TNF-α). Elevated TNF-α levels may indicate underlying inflammation within the male reproductive system and have been shown to impair fertilization potential by inducing oxidative stress via nitric oxide metabolites [94]. Nonetheless, this remains a working hypothesis in the absence of functional assays in sperm.

In necrosis-induced processes, DNase γ cooperates with DNase I. The first enzyme causes inter-nucleosomal DNA breakage, while DNase I is a secondary enzyme that causes random DNA digestion for its complete degradation [20]. Moreover, DNase γ operates in different organs to facilitate DNA removal [22] and is involved in different steps of sperm differentiation. Hence, this enzyme can be considered a good candidate to participate in the necrosis pathway, albeit its specific role remains unclear.

In humans, necrosis can occur in any sperm-producing individual; however, its prevalence increases with advancing age. A concomitant increase in necrosis and apoptosis, along with a decline in sperm motility, has been observed in aging males [95]. This effect becomes statistically significant as early as age 35 years, with distinct pathways implicated in sperm necrosis beyond the age of 40 years.

## 5. Sperm DNA Vulnerability During Assisted Reproductive Procedures

Following ejaculation and the subsequent application of in vitro techniques during ART, spermatozoa are removed from the protective environment of the male reproductive tract. Consequently, they become increasingly vulnerable to a spectrum of exogenous stressors including oxidative stress, fluctuations in temperature, variations in pH, osmotic conditions, and mechanical manipulation during laboratory handling. The deleterious impact of these factors is further compounded by the spermatozoa’s limited intrinsic capacity to counteract damage. As a result, all cellular components, particularly nuclear DNA, are highly susceptible to degradation.

This type of damage incurred during laboratory manipulation is classified as iatrogenic and is considered an unavoidable consequence of ART protocols [96,97,98,99]. Standard ART procedures, including centrifugation [98], cryopreservation [100], the reintegration of the first and second ejaculated fractions with seminal plasma during sperm collection [101,102], and the use of high sperm concentrations [103], may further exacerbate this baseline level of damage. Under such conditions, the SDF has been observed to increase progressively over time [8,104,105]. This time-dependent increase in SDF appears to be evolutionarily conserved, as documented in multiple animal species [106,107].

In clinical and diagnostic settings, it is essential to account for the temporal dynamics of SDF following ejaculation, as the increasing rate of DNA fragmentation can significantly influence the interpretation of results and the selection of appropriate fertilization strategies. Temporal variability in SDF assessments, often due to inconsistent intervals between sperm collection and analysis, can result in the misleading evaluation of DNA integrity. To enhance reproducibility and inter-laboratory comparability, it is recommended that SDF testing be standardized and performed 30 min post-ejaculation after complete liquefaction. Standardization is particularly crucial, given that the progression of SDF may follow exponential, linear, or logarithmic kinetics [8], with inter-individual variability in the extent of DNA fragmentation observed at identical time intervals.

For instance, a quantitative analysis of the increasing rate of SDF showed that, in sperm donors, SDF increases at an average rate of 8.3 units per hour during the first 4 h, 4.1 units per hour between 4 and 8 h, and 1.1 units per hour between 8 and 24 h post-ejaculation [8]. These rates may vary depending on the experimental conditions. Cryopreserved samples generally exhibit a more rapid SDF increase than fresh samples, particularly between 4 and 5 h post-thaw [108]. This finding has been supported by studies across different experimental models [105].

Figure 2 illustrates an example of inter-individual variation over time. Figure 2b shows the kinetic differences and the progression of SDF (scored at 0, 4, 8, and 24 h post-thaw) in cryopreserved sperm samples from three individuals. At baseline (t_0_), all samples exhibited low SDF values. By 4 h (t_4_), distinct fragmentation trajectories were observed: an individual with logarithmic kinetics showed a marginal increase (16% to 17%); another one displayed linear kinetics with moderate rise (12% to 30%), while the third case exhibited an exponential profile with a sharp escalation (20% to 59%) (Original information in [109]).

More sensitive methods, such as the two-tailed comet assay, have further confirmed significant increases in both SSBs and DSBs within 30 min of incubation at 37 °C in post-thawed cryopreserved sperm [110]. SDF progression kinetics within a given individual may differ under various experimental conditions, providing insights into sperm tolerance to environmental stressors [111].

A critical determinant of sperm resistance to iatrogenic stress is the protamination status of the chromatin. In humans, proper chromatin packaging requires the balanced incorporation of protamine 1 (P1) and protamine 2 (P2), ideally in a 1:1 ratio, in approximately half of the chromatin. The disruption of this balance, particularly a deficiency in P2, has been closely associated with increased SDF in ejaculated sperms [112,113]. From a kinetic perspective, aberrant P1/P2 ratios compromise chromatin integrity, rendering the DNA more susceptible to time-dependent damage [114].

The observed post-ejaculatory increase in sperm DNA fragmentation (SDF) raises critical questions regarding the underlying molecular mechanisms of DNA degradation. While mechanical stress and oxidative damage are well-established contributors, emerging evidence suggests that enzymatic activity, specifically from endogenous nucleases, may also play a pivotal role in sperm chromatin degradation, particularly in the context of assisted reproductive technologies (ART). Notably, the manipulation of semen samples during ART procedures may inadvertently activate these nucleases, thereby exacerbating DNA damage.

Experimental findings have shown that spermatozoa from mice and boars can internalize exogenous DNA, which in turn triggers the activation of a calcium-dependent endogenous nuclease. This nuclease initially targets the foreign DNA, but excessive intracellular DNA concentrations can lead to the autolytic degradation of the sperm genome [115]. This autodegradation mechanism is not observed in bovine sperm [116], indicating interspecies variability likely attributable to differences in chromatin compaction and nuclease accessibility. In individuals with spinal cord injuries, who frequently experience anejaculation, elevated levels of cell-free DNA have been reported in seminal plasma [117], along with increased sperm DNA fragmentation (SDF) [118,119]. In this context, it is tempting to speculate that, in these cases, internalization of cell-free DNA by spermatozoa would activate endogenous nucleases, which would contribute to the high levels of sperm DNA fragmentation observed in these patients.

In murine models, chromatin degradation by endogenous nucleases can also be induced by disrupting the sperm membrane with Triton X-100 in the presence of dithiothreitol (DTT), a disulfide bond-reducing agent. This nuclease activity is dependent on divalent cations, a characteristic feature of many nucleases, and can be inhibited by chelating agents such as EDTA and EGTA [120]. Similarly, in hamster spermatozoa, treatment with Triton X-100 and Mg^2+^ leads to chromatin degradation into 50 kb fragments, which correspond to the loop sizes of chromatin organized around protamine toroids [121,122]. These observations indicate that nuclease activity preferentially targets the less condensed linker regions of chromatin, analogous to the fragmentation patterns observed during apoptosis in somatic cells.

Endogenous nucleases have been identified in both murine and human sperm nuclei, with activity contingent on Ca^2+^ and Mg^2+^ availability [123]. A comparable Ca^2+^/Mg^2+^-dependent nuclease has also been characterized in the sperm nucleus of teleost fish, where it contributes to apoptotic-like chromatin degradation [124,125]. The conservation of such nuclear nucleases across diverse taxa suggests a fundamental physiological function, potentially related to fertilization.

Topoisomerase IIB has been implicated in the initial cleavage of sperm chromatin into 50 kb fragments, likely acting synergistically with exogenous nucleases. The removal of protamines from sperm chromatin renders it more susceptible to complete degradation by this enzymatic complex [126]. Additionally, nuclease activity is higher in seminal plasma than in epididymal fluid, and ejaculated sperm are more vulnerable to nuclease-mediated DNA damage than their epididymal counterparts [127].

Seminal plasma nuclease activity has been documented for several decades [128,129]. While seminal nucleases have been shown to induce DNA fragmentation in somatic cells [130], their in vivo role in SDF remains unclear. Notably, lower nuclease activity is generally observed in fertile donors compared to infertile patients, although no definitive correlation with SDF levels at ejaculation has been established [131].

Quantitative evidence of DNase activity in human seminal plasma remains limited. Nonetheless, Bartolomé et al. [132] demonstrated inter-individual variability in DNase activity across different ejaculates. Representative results are presented in Figure 3, where seminal plasma from 14 distinct ejaculates was used to digest genomic DNA.

The evolutionary conservation of seminal plasma nucleases among fish, birds, and mammals [133,134,135] implies a critical biological function despite their possible detrimental effects on sperm DNA integrity. One hypothesized role is in preventing the integration of exogenous DNA into sperm, thereby safeguarding genomic fidelity in the embryo. Another plausible function involves the degradation of compromised spermatozoa. In vitro data show that sperm membrane permeabilization activates endogenous nucleases and that complete DNA degradation requires the collaborative action of both endogenous and seminal plasma nucleases [126]. This mechanism may serve as a selective quality control process, ensuring that only membrane-intact, genetically competent sperm can achieve fertilization.

Additionally, upon entry into the female reproductive tract, sperm may elicit an inflammatory-like response leading to the recruitment of neutrophils and the formation of neutrophil extracellular traps (NETs), which are composed of DNA and associated proteins [136]. These NETs can entrap and neutralize spermatozoa [137], thereby impeding fertilization. Nucleases present in seminal plasma may facilitate sperm progression by degrading NETs [135,138].

While these nucleases may fulfil essential physiological roles, they could also pose risks by contributing to oocyte DNA damage during fertilization since, as just pointed out, seminal plasma nucleases can cause DNA breaks in cells in which the chromatin is organized around histone octamers, as is the case with oocytes. Importantly, follicular fluid has been shown to inhibit seminal plasma nuclease activity [132], which may explain why its addition to sperm culture media attenuates the temporal progression of SDF [139].

These findings carry important implications for ART protocols. First, since in vitro data have shown that the activation of endogenous nucleases is contingent upon membrane disruption, the gentle handling of sperm samples during procedures such as pipetting or centrifugation is advised. Second, due to the potentially deleterious effects of seminal plasma nucleases on sperm DNA integrity, it is advisable to separate sperm from seminal plasma as soon as possible after ejaculation. Experimental evidence indicates that prolonged exposure to seminal plasma is associated with a time-dependent increase in sperm DNA fragmentation [102]. Moreover, due to the protective effects of follicular fluid against nuclease-mediated DNA degradation, supplementing sperm culture media, particularly with follicular fluid from the female partner, may help mitigate iatrogenic damage and preserve sperm chromatin integrity.

## 6. Main Conclusions

The challenge in identifying the origin of sperm DNA breakage arises from the intricate biology of spermatogenesis, the multifactorial nature of DNA damage, and the inherent limitations of current diagnostic methodologies in accurately detecting such damage.

DNA fragmentation can originate from multiple sources and occur at various stages of sperm development. In the context of intracytoplasmic sperm injection, the putative DNA damage present in the ejaculate may be exacerbated by the in vitro manipulation of spermatozoa, which inevitably exposes the cells to iatrogenic stressors. Irrespective of its origin, the impact of SDF is particularly significant in spermatozoa due to their limited DNA repair capacity compared to somatic cells.

A considerable proportion of DNA fragmentation observed in the ejaculate is associated with physiological events during spermatogenesis, including programmed enzymatic activities involved in critical processes such as genetic recombination and the histone-to-protamine exchange. If these processes are not efficiently completed, spermatozoa containing fragmented DNA may persist and be present in the ejaculate. Additionally, exogenous factors such as oxidative stress, elevated temperatures, and environmental toxins can contribute to levels of DNA fragmentation that exceed physiological thresholds. In this context, intrinsic quality control mechanisms are responsible for identifying and eliminating spermatozoa harbouring defective DNA resulting from failed physiological processes or external stressors.

Apoptotic and necrotic pathways play essential roles in the removal of damaged germ cells and are active throughout various stages of gametogenesis. Each phase of gametogenesis is governed by distinct regulatory mechanisms related to cell death, often mediated by stage-specific enzymes or enzymatic pathways. Some of these enzymatic pathways or enzymes, which involve proteins with deoxyribonuclease activity, contribute to the generation and/or persistence of DNA damage detectable in the ejaculate.

The presence and interindividual variability of DNase activity in seminal plasma is particularly relevant in the context of ART, as these enzymatic activities may serve as a post-testicular source of additional DNA fragmentation in spermatozoa that otherwise exhibited an intact DNA structure without detectable breaks in the ejaculate. The potential involvement of DNases, such as DNase I and II and other enzymes like SPO11 or topoisomerases, which may be released during cellular degradation processes at various stages of gametogenesis, warrants further investigation to clarify their physiological contribution to sperm DNA fragmentation, particularly in relation to iatrogenic damage associated with ART procedures.

## Figures and Tables

**Figure 1 ijms-26-06789-f001:**
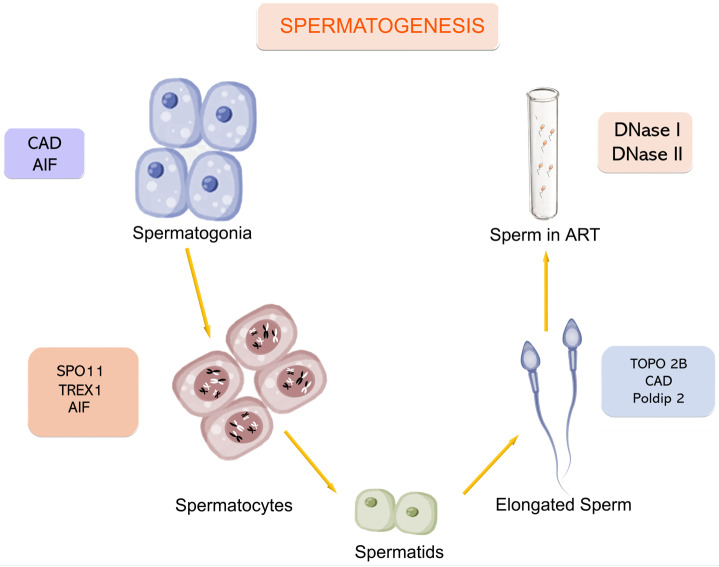
Main enzymes involved in DNA breakage and the stages of spermatogenesis in which they intervene. CAD: Caspase-Activated DNase; AIF: Apoptosis-Inducing Factor; SPO11: Initiator of meiotic double strand breaks; TREX1: Three Prime Repair Exonuclease 1; TOPO2B: DNA Topoisomerase II Beta; Poldip2: DNA Polymerase delta-interacting protein 2; DNase I: Deoxyribonuclease I; DNase II: Deoxyribonuclease II.

**Figure 2 ijms-26-06789-f002:**
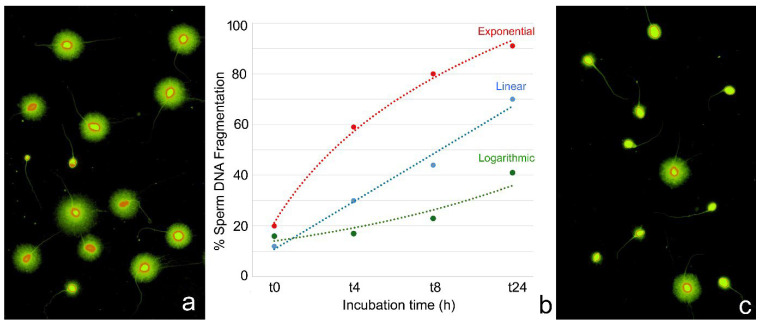
Sperm DNA fragmentation (SDF) variation over time. Representative images of SDF levels are shown at t_0_ ((**a**): 30 min post-ejaculation, low SDF level) and t_24_ ((**c**): after 24 h of sperm incubation, high SDF level). Intact chromatin is indicated by large halos surrounding the sperm nuclei, whereas fragmented DNA presents as either small or absent halos. (**b**) Three illustrative cases with comparable baseline SDF levels (t_0_) but exhibiting distinct temporal progression patterns: logarithmic (green line), linear (blue line), and exponential (red line) increases. Notably, while SDF levels appear similar at t_0_, marked differences emerge at t_4_, t_8_, and t_24_. Technical details in [8].

**Figure 3 ijms-26-06789-f003:**
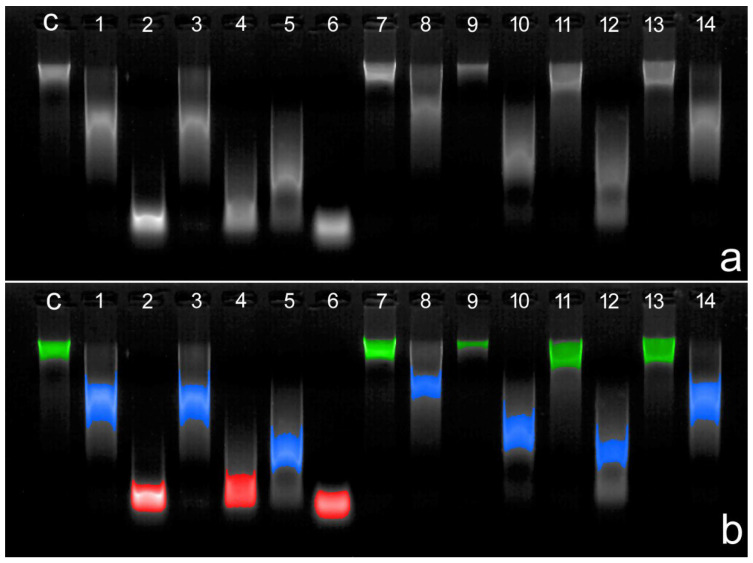
DNase activity in seminal plasma as assessed by gel electrophoresis of genomic DNA incubated with seminal plasma from 14 different human ejaculates. DNA samples were incubated with seminal plasma for 30 min at 37 °C. (**a**) Original gel image. (**b**) Pseudocolored image to enhance inter-individual variation in DNA cleavage. Regions of low, moderate, and high DNase activity are indicated in green, blue, and red, respectively. C: Control: DNA incubated only with dH_2_O. Technical details in [132].

**Table 1 ijms-26-06789-t001:** A summary of the distinct proteins intervening in DNA breakages throughout sperm development. See also the legend of Figure 1 for enzyme acronyms.

Enzyme	Stage	Localization	Function/Role	Notes *
AIF	Spermatogonia and primary spermatocytes	Mitochondria → Nucleus	Caspase-independent apoptosis, large-scale DNA fragmentation (50 kb–1 Mb)	Low expression in mature sperm [10,11]
TREX1	Early spermatogenesis	Cytosol/Nucleus of Sertoli cells	Degrading DNA fragments post-apoptosis. Preventing inflammation	Clearing DNA debris from CAD activity [12]
SPO11	Early meiosis I	Nucleus	Introducing programmed DSBs to initiate meiotic recombination	Required for synapsis [13]
EndoG	Late spermatogenesis (post-meiotic)	Mitochondria → Nucleus	Cleaving nuclear and mitochondrial DNA during apoptosis	Role in sperm is debated [14,15]
Poldip2	Spermiogenesis	Mitochondria	Clearing mitochondrial DNA in elongating spermatids	Loss leading to genome fragmentation [16]
TOPO2B	Spermiogenesis	Nucleus	Inducing DSBs during histone-to-protamine exchange	Working with torsional stress [17]
CAD	Early spermatogenesis and spermiogenesis	Nucleus	Cleaving chromosomal DNA into nucleosomal fragments during apoptosis	Activated by Caspase-3/-7 [18,19]
DNase I	Ejaculated sperm	Extracellular and lysosomal	Random DNA cleavage in necrotic cells	Ca^2+^ and Mg^2+^ required [20]
DNase II	Ejaculated sperm	Lysosomes	Digesting apoptotic/necrotic DNA under acidic pH in the female tract	Independent of divalent cations [21]
DNase γ	Possibly across all stages	Unclear-multi-organ	Fragmenting internucleosomal DNA. Cooperating with DNase I especially during necrosis	Suggested but not confirmed in sperm [20,22]

* Corresponding reference is indicated in square brackets.

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
