# Peer review of "Hurdles of Sperm Success: Exploring the Role of DNases"

_ijms, 2025, doi:10.3390/ijms26146789_

Round 1
Reviewer 1 Report
Comments and Suggestions for Authors
Gosalvez et al. provide a comprehensive review of the impact of evaluating DNA fragmentation in the reproductive process, with a particular focus on its relevance in assisted reproduction techniques. The review also emphasizes an under-researched aspect: the role of DNases from the proliferation of the male gonial cells to spermatozoa. As well as, provides information on the impact of assisted reproduction techniques and cryopreservation on sperm DNA. The use of cryopreservation is becoming increasingly prevalent in these programs. This review constitutes a significant contribution to the fields of reproductive medicine and the basic sciences underlying DNA fragmentation.
Please analyze the number of self-citations in comparison to the total number of citations.
Author Response
Response to Reviewer 1 & Comments
Thank you very much for taking the time to review this manuscript. We have carefully considered your comments. Please find our detailed responses below.
Point-by-Point Response to Comments and Suggestions for Authors
Comment 1: Please analyze the number of self-citations in comparison to the total number of citations.
Response 1: Thank you for pointing this out. We have reviewed the number of self-citations and found that there are 20 self-citations out of a total of 139 references. Most of these refer to our previous work on the dynamics of DNA degradation under ART, as well as to experiments on the role of DNases that were originally developed and published by our group or in collaboration with others. DNA stability after ejaculation is a critical aspect that deserves further investigation, and we have taken the opportunity in this paper to emphasize its importance.
Reviewer 2 Report
Comments and Suggestions for Authors
This review addresses the important issue of sperm DNA fragmentation, focusing on a less commonly explored mechanism — the role of deoxyribonucleases (DNases). By highlighting how these enzymes may contribute to DNA damage at various stages of sperm development, from spermatogonia to ejaculated sperm, and during assisted reproductive procedures, the authors bring a valuable perspective to the field. The review is well written and thoughtfully structure. I have no major concerns with respect to the overall quality of the paper, I do have only a couple of minor comments:
- I suggest to replace the term “gonial cell” for spermatogonia - which is the more widely recognized and commonly used term
- Page 4, line 122 - the sentence may unintentionally imply that apoptosis itself functions as the checkpoint mechanism. It could be worth slightly rephrasing to clarify that apoptosis is the result of checkpoint activation, rather than the checkpoint itself.
- Page 5, lines 141-152 - I noticed that this section is written in italics. Is this a direct quotation or was the formatting intended for emphasis?
- Page 9, line 359 - since spermiogenesis is the final stage of spermatogenesis, listing them separately may give the impression that they are distinct processes.
- Page 10, line 369 - apoptosis and necrosis are considered distinct and independent forms of cell death. While it is possible that necrosis follows incomplete or abortive apoptosis (as in secondary necrosis), this is not typically described as a sequential "phase" following apoptosis in standard germ cell development.
Author Response
Response to Reviewer 2 & Comments
Thank you very much for taking the time to review this manuscript. Please find our detailed responses below and the corresponding corrections highlighted in the re-submitted manuscript.
- Point-by-point response to Comments and Suggestions for Authors
Comment 1: I suggest to replace the term “gonial cell” for spermatogonia - which is the more widely recognized and commonly used term
Response 1: Thank you for pointing this out. We have thoroughly reviewed the text and incorporated the changes where appropriate. The specific revisions can be found in the following sections of the original manuscript: Line 32, Line 62, Line 157 and Line 371.
Comment 2: Page 4, line 122 - the sentence may unintentionally imply that apoptosis itself functions as the checkpoint mechanism. It could be worth slightly rephrasing to clarify that apoptosis is the result of checkpoint activation, rather than the checkpoint itself.
Response 2: We agree with this comment. We have rewritten the sentence to clarify this point in Line 123-124.
Comments 3: Page 5, lines 141-152 - I noticed that this section is written in italics. Is this a direct quotation or was the formatting intended for emphasis?
Response 3: Thank you very much for this remark. This was a formatting error. This paragraph should not have appeared in italics. The issue has been corrected in the revised version of the manuscript. Lines 142-155.
Comments 4: Page 9, line 359 - since spermiogenesis is the final stage of spermatogenesis, listing them separately may give the impression that they are distinct processes.
Response 4: We agree with this comment and have rephrased the corresponding sentence in Line 367-368. Accordingly with this change we have also modified Figure 1.
Comments 5: Page 10, line 369 - apoptosis and necrosis are considered distinct and independent forms of cell death. While it is possible that necrosis follows incomplete or abortive apoptosis (as in secondary necrosis), this is not typically described as a sequential "phase" following apoptosis in standard germ cell development.
Response 5: We agree with this comment and have rephrased the corresponding sentence in Lines 376-377.
Reviewer 3 Report
Comments and Suggestions for Authors
This is a comprehensive and well-written review article that addresses a biologically and clinically relevant topic: the role of deoxyribonucleases (DNases) in sperm DNA fragmentation (SDF) throughout male gametogenesis and during post-ejaculatory stages, including assisted reproductive technologies (ART). The manuscript provides an in-depth overview of molecular pathways contributing to DNA integrity loss, integrating physiological, pathological, and iatrogenic mechanisms.
DNase involvement during gametogenesis: this section is detailed and technically robust, although at times they favor description over interpretation. Table 1 is highly informative, and it might be strengthened by more clearly distinguishing between physiological and pathological roles of the enzymes listed. Likewise, the flow of information in these early sections could be improved by adjusting paragraph transitions and ensuring terminological consistency when referring to DNA cleavage pathways. In some cases, the roles of apoptotic versus necrotic DNases appear blurred, and further clarity would be helpful for readers less familiar with these mechanisms.
Apoptosis during spermatogenesis: The notion of abortive apoptosis is addressed convincingly, although a brief concluding paragraph synthesizing its potential contribution to persistent SDF in the ejaculate would be welcome. Regarding the controversial presence of apoptosis in ejaculated sperm, the manuscript does a good job of weighing evidence, but the interpretations could be phrased more cautiously in the absence of functional validation.
The necrosis section adds valuable insight, particularly in the context of post-ejaculatory sperm clearance by the female immune system. However, the distinction between apoptosis and necrosis could be reinforced, especially when both markers co-exist. The idea that necrosis may play a predominant role under conditions of oxidative stress or inflammation is plausible and aligns with broader findings in the literature, but it would be worth explicitly noting that this remains a working hypothesis in the absence of functional assays in sperm.
The section on ART-induced damage is timely and highly relevant. The kinetic analysis of SDF post-ejaculation is compelling, and the authors do well in highlighting inter-individual variability. The suggestion that nucleases may be inadvertently activated during sperm handling is particularly interesting but should be framed as a hypothesis supported by indirect evidence. The discussion on the protective role of follicular fluid is also a strong point and could be emphasized more as a practical consideration for ART protocols.
Throughout the text, there are minor issues with consistency in terminology. Terms like “chromatin degradation,” “DNA damage,” and “DNA fragmentation” are sometimes used interchangeably, and could be more precisely defined depending on the context. Enzyme names should also be standardized (e.g., DNase I, DNase II, DNase γ), and abbreviations should be defined clearly at first use. Some sentences are overly long and could be simplified for clarity, especially in the Introduction and Discussion. Figures and tables are generally clear and well designed, though it is important to ensure that all abbreviations are explained in the legends and that image resolution meets journal requirements.
Author Response
Response to Reviewer 3 & Comments
Thank you very much for taking the time to review our manuscript and for your valuable suggestions. Please find our detailed responses below, with the corresponding corrections highlighted in the revised manuscript.
Point-by-point response to Comments and Suggestions for Authors
Comment 1: DNase involvement during gametogenesis: this section is detailed and technically robust, although at times they favor description over interpretation. Table 1 is highly informative, and it might be strengthened by more clearly distinguishing between physiological and pathological roles of the enzymes listed. Likewise, the flow of information in these early sections could be improved by adjusting paragraph transitions and ensuring terminological consistency when referring to DNA cleavage pathways. In some cases, the roles of apoptotic versus necrotic DNases appear blurred, and further clarity would be helpful for readers less familiar with these mechanisms.
Response 1:
Table 1 is highly informative, and it might be strengthened by more clearly distinguishing between physiological and pathological roles of the enzymes listed
Table 1 summarizes the enzymes mentioned in the text, all of which function under physiological conditions. In other words, these enzymes are naturally active during sperm formation. In fact, we have deliberately avoided discussing most pathological conditions to emphasize the importance of these enzymes in successful sperm development. Furthermore, we have revised the content of Table 1 and Figure 1 to ensure their consistency.
Likewise, the flow of information in these early sections could be improved by adjusting paragraph transitions and ensuring terminological consistency when referring to DNA cleavage pathways
We have separated some paragraphs to make the text more fluid.
In some cases, the roles of apoptotic versus necrotic DNases appear blurred, and further clarity would be helpful for readers less familiar with these mechanisms.
We agree that the role of DNase activity in both processes is somewhat unclear. We have attempted to clarify this point in the relevant sections, based on the current state of knowledge.
Comment 2: Apoptosis during spermatogenesis: The notion of abortive apoptosis is addressed convincingly, although a brief concluding paragraph synthesizing its potential contribution to persistent SDF in the ejaculate would be welcome. Regarding the controversial presence of apoptosis in ejaculated sperm, the manuscript does a good job of weighing evidence, but the interpretations could be phrased more cautiously in the absence of functional validation.
Response 2:
Apoptosis during spermatogenesis: The notion of abortive apoptosis is addressed convincingly, although a brief concluding paragraph synthesizing its potential contribution to persistent SDF in the ejaculate would be welcome.
We have added a sentence for reinforcing this idea (Lines 153-155).
Regarding the controversial presence of apoptosis in ejaculated sperm, the manuscript does a good job of weighing evidence, but the interpretations could be phrased more cautiously in the absence of functional validation.
Thanks for this observation. We feel that this cautious view is already clearly expressed on Lines 278-282.
Comment 3: The necrosis section adds valuable insight, particularly in the context of post-ejaculatory sperm clearance by the female immune system. However, the distinction between apoptosis and necrosis could be reinforced, especially when both markers co-exist. The idea that necrosis may play a predominant role under conditions of oxidative stress or inflammation is plausible and aligns with broader findings in the literature, but it would be worth explicitly noting that this remains a working hypothesis in the absence of functional assays in sperm.
Response 3:
We have added a new statement noting that necrosis may play a role under these circumstances, while emphasizing the need for functional assays to confirm this (Lines 402 – 406).
Comments 4: The section on ART-induced damage is timely and highly relevant. The kinetic analysis of SDF post-ejaculation is compelling, and the authors do well in highlighting inter-individual variability. The suggestion that nucleases may be inadvertently activated during sperm handling is particularly interesting but should be framed as a hypothesis supported by indirect evidence. The discussion on the protective role of follicular fluid is also a strong point and could be emphasized more as a practical consideration for ART protocols.
Response 4:
The suggestion that nucleases may be inadvertently activated during sperm handling is particularly interesting but should be framed as a hypothesis supported by indirect evidence.
We have added three different sentences throughout the text of this section. They are now included in the following Lines: 498-501. Lines: 556-559. Lines 562-564.
The discussion on the protective role of follicular fluid is also a strong point and could be emphasized more as a practical consideration for ART protocols.
After reviewing the original text, we believe that this idea is already addressed in Lines 569-572.
Comments 5: Throughout the text, there are minor issues with consistency in terminology. Terms like “chromatin degradation,” “DNA damage,” and “DNA fragmentation” are sometimes used interchangeably, and could be more precisely defined depending on the context. Enzyme names should also be standardized (e.g., DNase I, DNase II, DNase γ), and abbreviations should be defined clearly at first use. Some sentences are overly long and could be simplified for clarity, especially in the Introduction and Discussion. Figures and tables are generally clear and well designed, though it is important to ensure that all abbreviations are explained in the legends and that image resolution meets journal requirements.
Response 5:
Throughout the text, there are minor issues with consistency in terminology. Terms like “chromatin degradation,” “DNA damage,” and “DNA fragmentation” are sometimes used interchangeably, and could be more precisely defined depending on the context.
There are three places (see below) where we modified the expression. We thank you for the warning, but we must say that care has been taken to use 'breakage or damage' to describe the molecular event, and 'degradation or fragmentation' as the consequence of the former. We have modified these terms in Line 314, Line 408 and Line 574.
Enzyme names should also be standardized (e.g., DNase I, DNase II, DNase γ), and abbreviations should be defined clearly at first use.
Each enzyme has been reviewed throughout the text, and abbreviations have been properly defined both in the main text and in the figure legends. For instance, see Lines: 149, 150, 186, 196, 213, 215, 220, 221, 242, 245, 253, 317 and 320.
Some sentences are overly long and could be simplified for clarity, especially in the Introduction and Discussion.
We have divided some conceptual paragraphs upon realizing that the concepts they addressed were distinct enough to be presented separately.
Figures and tables are generally clear and well designed, though it is important to ensure that all abbreviations are explained in the legends and that image resolution meets journal requirements.
Thank you very much for this positive comment. With respect to the resolution of each image, they meet journal requirements.